# Exploring the State and Action Space in Reinforcement Learning with Infinite-Dimensional Confidence Balls

## Abstract

Reinforcement Learning (RL) is a powerful tool for solving complex decision-making problems. However, existing RL approaches suffer from the curse of dimensionality when dealing with large or continuous state and action spaces. This paper introduces a non-parametric online RL algorithm called RKHS-RL that overcomes these challenges by utilizing reproducing kernels and the RKHS-embedding assumption. The proposed algorithm can handle both finite and infinite state and action spaces, as well as nonlinear relationships in transition probabilities. The RKHS-RL algorithm estimates the transition core using ridge regression and balances exploration and exploitation through infinite-dimensional confidence balls. The paper provides theoretical guarantees, demonstrating that RKHS-RL achieves a sublinear regret bound of $\tilde{\mathcal{O}}(H\sqrt{T})$, where $T$ denotes the time step of the algorithm and $H$ represents the horizon of the Markov Decision Process (MDP), making it an effective approach for RL problems.

## 1 Introduction

Reinforcement Learning (RL) is a subfield of machine learning that aims to enable an agent to learn how to make sequential decisions in an environment to maximize a cumulative reward. RL is inspired by the concept of trial and error, which is how humans and animals learn from their interactions with the world. The ability of RL to learn from interactions with the environment and make decisions without explicit supervision has attracted significant attention in diverse domains, including game AI (Silver et al., 2016), robotics (Ibarz et al., 2021), healthcare (Esteva et al., 2019) and autonomous driving (Kiran et al., 2021). Its ability to learn from interactions with the environment and make decisions without explicit supervision makes RL a powerful tool for solving complex decision-making problems.

The Markov decision process (MDP) provides a formal framework for studying reinforcement learning (Van Otterlo & Wiering, 2012; Puterman, 1990; Hu & Yue, 2007). In MDPs, an agent interacts with the environment in discrete time steps. At each time step, the agent observes the current state of the environment and selects an action to perform. The environment then transitions to a new state based on the current state and the action taken by the agent. The agent's goal is to learn an optimal policy that maximizes the expected cumulative reward over time, which is represented by the value function.

Researchers propose several assumptions, such as tabular settings, linear MDPs, and parametric MDPs, to simplify the RL model and solve the problem. In the tabular setting, where the state and action spaces ($\mathcal{S}$ and $\mathcal{A}$) are small and finite, a nearly sublinear regret algorithm is available that achieves a regret bound $\tilde{\mathcal{O}}(H^{\frac{3}{2}}\sqrt{|\mathcal{S}||\mathcal{A}|T})$ (Auer et al., 2008; Azar et al., 2017). Here, $T$ is the time step of the algorithm, $H$ is the horizon of the MDP, and $\tilde{\mathcal{O}}(\cdot)$ hides polylog factors of the input. The algorithm balances exploration and exploitation by defining a reward bonus in each step. However, it suffers from the curse of dimensionality when $\mathcal{S}$ and $\mathcal{A}$ are large or infinite.

In the case of infinite and discrete state and action spaces, linear parameterized MDP adds a linear structure to both the reward and transition probability. In linear MDPs, value functions are a linear combination of features extracted from the state-action space, which reduces the computational burden required to learn the optimal policy. Linear methods can be solved analytically, providing

a more efficient approximation to the optimal value function. For linear MDPs, algorithms exist that achieve a regret bound of $\tilde{\mathcal{O}}(H^{\frac{3}{2}}\sqrt{d^3 T})$ (Jiang et al., 2017; Yang & Wang, 2020; Jin et al., 2020), where $d$ is the dimension of the feature space. However, linear MDPs have limited representational power and cannot generalize well in high-dimensional spaces. They can only capture linear relationships between features, which may not be sufficient for modeling complex interactions and dynamics in continuous spaces.

When the state and action spaces are continuous, there are several approaches to study continuous MDPs. The first is the parametric method, which includes studies on parametric MDPs with exponential families and algorithms that achieve a regret bound of $\tilde{\mathcal{O}}(dH\sqrt{T})$ (Chowdhury et al., 2021). There are also studies on RL with parameterized actions (Masson et al., 2016). These approaches allow for generalization across the continuous state space. However, selecting an appropriate parametric function and architecture can be challenging, and they still suffer from the curse of dimensionality in high-dimensional state spaces. The second approach is the kernel method, which includes studies on kernel-based temporal difference learning (Ormoneit & Sen, 2002) and kernel-based least-squares policy iteration (Xu et al., 2007). Kernel methods allow for nonlinear approximation of the value function by implicitly mapping state-action pairs to a high-dimensional feature space, capturing complex relationships and dependencies. However, kernel methods also suffer from the curse of dimensionality, particularly when dealing with high-dimensional state and action spaces. When the number of state-action pairs grows, the kernel matrix can become large, leading to increased computational requirements and potential overfitting. So we want to use the reproducing property of RKHS to solve the problem. The reproducing property of RKHS allows for a natural way to evaluate and compare functions. So it's meaningful to use RKHS to study RL. Finally, approaches using neural networks also exist (Mnih et al., 2015). But understanding the relationship between input features and output predictions in neural networks can be difficult. Reproducing kernel Hilbert space (RKHS) has advantages over kernels.

Reproducing kernel Hilbert spaces (RKHS) are Hilbert spaces equipped with a reproducing kernel that have proven useful in regression, classification, and clustering problems (Berlinet & Thomas-Agnan, 2011; Gu & Gu, 2013; Shawe-Taylor et al., 2004; Wainwright, 2019). RKHS offers additional properties and advantages over kernels, making it a powerful tool for function approximation and learning tasks. This framework is versatile and reliable for solving various machine learning problems, due to its completeness, reproducing property, well-defined inner product, function approximation capabilities, convergence, and stability. Therefore, studying RL problems in reproducing kernel Hilbert spaces is meaningful.

There is a growing body of studies that combine RKHS and RL. There are lines of work that give an RKHS version of traditional RL algorithms (Robards et al.; Valko et al., 2013). Robards et al. studies RKHS temporal difference learning and Valko et al. (2013) kernelizes the linear UCB algorithm. However, their algorithm relies on the finiteness of the state space $\mathcal{S}$, which is overcome in our paper. There are some works that propose some general frameworks to study RL with RKHS (Du et al., 2021; Long & Han, 2021). Du et al. (2021) solved RKHS linear MDPs and RKHS mixture MDPs by creating the "Bilinear Classes". However, "Bilinear Classes" is hard to verify in practice and our paper only needs the transition probability to be smooth enough. Long & Han (2021) defines a quantity called perturbational complexity by distribution mismatch and use this quantity to measure the upper bound of two algorithms (fitted reward and fitted Q-iteration). However, more algorithms are not given. Chowdhury & Gopalan (2019) gives an algorithm for learning MDPs with mean transition dynamics and reward structure assumed to belong to appropriate RKHSs. The algorithm achieves regret bound $\tilde{\mathcal{O}}\big((\gamma_T(R) + \gamma_{mT}(P))\sqrt{T}\big)$, where $\gamma_t(\cdot)$ roughly represents the maximum information gain about the unknown dynamics. But this paper assumes value functions are Lipschitz continuous, which is not assumed in our paper.

This paper aims to address the curse of dimensionality, the strict assumptions on RL models, and the lack of theoretical guarantees. To overcome these challenges, we propose modeling RL problems using the RKHS-embedding assumption, which exhibits generalization abilities and can handle complex environments. Specifically, we utilize the following equation to model RL problems:

$$P(\widetilde{s}|s,a) = \left\langle \iint_{\mathcal{S}\times\mathcal{A}} \Phi_{s,a}(x,z) M^*((x,z),\cdot) dx dz, \Psi_{\widetilde{s}}(\cdot) \right\rangle_{\mathcal{H}_2},$$

Here, the reproducing kernel $\Phi_{s,a}(\cdot) = \mathcal{K}_1(\cdot, (s,a))$, $\Psi_{\widetilde{s}}(\cdot) = \mathcal{K}_2(\cdot, \widetilde{s})$ are given as a priori, and $M^*$ is the transition core. Instead of naively assuming that the transition probability $P$ belongs to

an RKHS, we actually assume a more delicate structure of $P$ which is that $P$ has a decomposable structure. Only on this assumption can we derive an efficient algorithm to conduct RL but still make the model quite general as long as $P$ is smooth enough. Unlike tabular MDPs, our proposed model can handle both finite and infinite state and action spaces. Unlike linear MDPs and feature methods, our model can handle nonlinear relationships in transition probability. In contrast to parametric approaches, our model does not require selecting an appropriate parametric function. Unlike neural network methods, our model has a mathematical interpretation. Our non-parametric model utilizing the RKHS-embedding assumption can capture intricate nonlinear relationships that may not be easily captured by traditional linear methods, enabling more expressive and flexible modeling of complex data patterns. Moreover, the reproducing property of RKHS plays an essential role in embedding learning, guaranteeing that the inner product between a learned embedding and a function in the RKHS evaluates the function at the corresponding data point.

In this paper, we propose a non-parametric online RL algorithm called RKHS-RL to solve the model, which utilizes reproducing kernels to learn the transition model. To explore the state and action space, RKHS-RL estimates the transition core via ridge regression and balances exploration and exploitation by constructing a confidence ball. Previous work by (Yang & Wang, 2020) proposed a finite-dimensional confidence ball to solve feature-embedding RL problems using mathematical techniques related to linear algebra. However, the non-parametric and infinite-dimensional nature of RKHS-embedding RL problems poses significant challenges. To address this issue, we propose an infinite dimensional confidence ball using mathematical techniques related to functional analysis. Our confidence ball is essentially a sequence of confidence regions that converge to the transition core as the time step increases, which helps the agent balance RL exploration and exploitation. Additionally, we creatively generalize matrix multiplication into Hilbert space throughout our analysis.

We demonstrate that RKHS-RL achieves the regret bound of $\tilde{\mathcal{O}}(H\sqrt{T})$, ensuring that it can learn the optimal policy when $T$ is sufficiently large. Our regret bound does not rely on the dimension of state and action spaces, unlike the regret bound $\tilde{\mathcal{O}}(H^{\frac{3}{2}}\sqrt{|\mathcal{S}||\mathcal{A}|T})$ for tabular MDPs (Auer et al., 2008; Azar et al., 2017). Nor does it rely on the number of features, unlike the regret bound $\tilde{\mathcal{O}}(H^{\frac{3}{2}}\sqrt{d^3 T})$ for Linear MDPs and feature methods (Jiang et al., 2017; Yang & Wang, 2020; Jin et al., 2020). These advantages enable RKHS-RL to avoid the curse of dimensionality. Note that for linear bandit, a special case of RL, the regret lower bound is $\tilde{\Omega}(d\sqrt{T})$ (Dani et al., 2008). Our regret bound matches the lower bound up to polylog factors in $T$. To our best knowledge, we provide the first regret bound that is simultaneously near-optimal in the time $T$, polynomial in the planning horizon $H$ and independent of the feature dimension $d$.

**Contributions.** We summarize our contributions as follows:

- Model: We formulate online RL problems by RKHS-embedding assumption, which exhibits abilities of generalization and dealing with complex environments.

- Method: Our algorithm, RKHS-RL, provides a non-parametric way to learn the transition model using given reproducing kernels. It balances exploration and exploitation by constructing an infinite dimensional confidence ball.

- Theory: RKHS-RL achieves the regret bound $\tilde{\mathcal{O}}(H\sqrt{T})$. It is near-optimal in the time $T$, polynomial in the planning horizon $H$ and independent of the feature dimension $d$.

## 1.1 RELATED LITERATURE

There is an extensive body of work on solving MDPs under various assumptions. In the tabular setting, where both the state space $\mathcal{S}$ and action space $\mathcal{A}$ are finite, several methods achieve sublinear regret for $H$-horizon episodic RL. For instance, Auer et al. (2008) and Azar et al. (2017) achieve sublinear regret of $\tilde{\mathcal{O}}(H^{\frac{3}{2}}\sqrt{|\mathcal{S}||\mathcal{A}|T})$, while Dann et al. (2019) and Jin et al. (2018) achieve regret asymptotically of $\mathcal{O}(\sqrt{H|\mathcal{S}||\mathcal{A}|T})$. For linear MDPs, it has been observed that such assumptions can lead to statistically efficient algorithms due to their low Bellman rank (Jiang et al., 2017). The first algorithm that achieves statistically and computationally efficient learning for linear MDPs achieves regret bound of $\mathcal{O}(H^2 d \log T \sqrt{T})$ (Yang & Wang, 2020), where $d$ represents the feature space dimension. A simplified version of this model and algorithm achieves regret bounds of $\tilde{\mathcal{O}}(H^{\frac{3}{2}}\sqrt{d^3 T})$ (Jin et al., 2020). For parametric methods in RL problems, there exist algorithms that

achieve regret bounds of $\tilde{\mathcal{O}}(dH\sqrt{T})$ (Chowdhury et al., 2021) in parametric MDPs with exponential families. Studies have also been conducted on RL with parameterized actions (Masson et al., 2016). In kernel methods for RL problems, there are studies on kernel-based temporal difference learning (Ormoneit & Sen, 2002) and kernel-based least-squares policy iteration (Xu et al., 2007).

Reproducing kernel Hilbert spaces have also been widely studied, with studies on the least square regression model in an RKHS for regression (Rosipal & Trejo, 2001), the hard and soft classification model by RKHS for classification (Wahba, 2002), and the clustering algorithm using RKHS for clustering (Paiva et al., 2009). There is also a line of work on RKHS embedding. For instance (Fukumizu et al., 2009; Sriperumbudur et al., 2010; Balasubramanian et al., 2013). For value function approximation, there are methods based on RKHS for estimating the value function of an infinite-horizon discounted Markov reward process (Duan et al., 2021).

For the combination of RKHS and RL, there is also a line of work. Robards et al. studies RKHS temporal difference learning and Valko et al. (2013) kernelizes the linear UCB algorithm. Du et al. (2021); Long & Han (2021) propose some general frameworks to study RL with RKHS. Du et al. (2021) solved RKHS linear MDPs and RKHS mixture MDPs by creating the "Bilinear Classes". Long & Han (2021) defines a quantity called perturbational complexity by distribution mismatch and use this quantity to measure the upper bound of two algorithms (fitted reward and fitted Q-iteration). Chowdhury & Gopalan (2019) gives an algorithm for learning MDPs with mean transition dynamics and reward structure assumed to belong to appropriate RKHSs.

## 2 BACKGROUND AND PROBLEM SET-UP

In this section, we provide background before formulating the algorithm to be analyzed. Section 2.1 introduces basic concepts in a Markov Decision Process (MDP). Section 2.2 is devoted to background on reproducing kernel Hilbert space (RKHS).

### 2.1 PROBLEM FORMULATION

In this paper, we consider a finite-horizon Markov Decision Process (MDP) denoted by a tuple $\mathcal{M} = (\mathcal{S}, \mathcal{A}, P, \{r_h\}, H, s_0)$, where $\mathcal{S}$ and $\mathcal{A}$ are continuous sets of states and actions, respectively. At any state $s \in \mathcal{S}$, the agent can select an action $a \in \mathcal{A}$. The agent receives an immediate reward $r_h(s, a) \in [0, 1]$ and transitions to the next state $s' \in \mathcal{S}$ with probability $P(s'|s, a)$ after taking action $a$ at state $s$. After $H$ steps, the process restarts at an initial state $s_0$. In an MDP, the principal goal is to find a policy $\pi : \mathcal{S} \times [H] \to \mathcal{A}$ that maximizes the long-term expected reward, starting from a given state $s$ and stage $h \in [H]$. For a policy $\pi$, a state $s$, and $h \in [H]$, we define the value function $V_h^\pi : \mathcal{S} \to \mathbb{R}$ as

$$V_h^\pi(s) := \mathbb{E}\left[\sum_{t=h}^{H} r_t(s_t, a_t)|\pi, s_h = s\right].$$

A policy $\pi^*$ is optimal if it attains the maximal possible value at every state $s$ and every stage $h$. We denote $V^*$ as the optimal value function. Also, we denote the optimal action-value function as

$$Q_H^*(s, a) = r_H(s, a),$$
$$Q_h^*(s, a) = r_h(s, a) + \mathbb{E}_{s' \sim P(\cdot|s,a)} V_{h+1}^*(s'), \qquad \forall h \in [H-1],$$

In online Reinforcement Learning (RL) settings, the agent interacts with the environment episodically, where an episode starts at $s_0$ and lasts for $H$ steps. Let $n$ denote the current episode number and $t = (n-1)H + h$ denote the current step. The performance of a learning algorithm is evaluated by its regret, which is defined as the difference between the cumulative reward of the optimal policy and the cumulative reward of the learning algorithm. Here, we adopt the following definition of regret.

**Definition 1.** *In an MDP $\mathcal{M} = (\mathcal{S}, \mathcal{A}, \mathcal{P}, \{r_h\}, H, s_0)$, the regret for an algorithm $\mathcal{K}$ at step $T = NH$ is defined as*

$$\text{Regret}(T) = \mathbb{E}_{\mathcal{K}}\left[\sum_{n=1}^{N}\left(V^*(s_0) - \sum_{h=1}^{H} r_h(s_{n,h}, a_{n,h})\right)\right],$$

*where $\mathbb{E}_{\mathcal{K}}$ is expectation taken over the random path of states under algorithm $\mathcal{K}$.*

In this paper, we study the problem where the transition function $P$ can be embedded in a given reproducing kernel Hilbert space. Let $0 < c < C < \infty$ be positive parameters from case to case.

**Remark.** In our paper, we need to know the immediate reward $r_h(s, a)$ after playing $a$ at $s$. This is in fact without loss of generality because learning about the environment $P$ is much harder than learning about $r$. In the case if $r$ is unknown and satisfies certain conditions, we can extend our algorithm by adding a step of optimistic reward estimation like in LinUCB. There are also works that have the same assumptions on the reward.(Yang & Wang, 2020; Agrawal & Jia, 2017; Azar et al., 2017)

## 2.2 MDP Model with Reproducing Kernel Hilbert Space

Reproducing kernel Hilbert space (RKHS) provides a fertile ground for developing non-parametric estimators. An RKHS is a particular type of Hilbert space of real-valued functions $f$ with domain $\mathcal{X}$. As a Hilbert space, the RKHS has an inner product $\langle f, g \rangle_{\mathcal{H}}$ along with the associated norm $||f||_{\mathcal{H}}$. There exists a symmetric kernel function $\mathcal{K} : \mathcal{X} \times \mathcal{X} \to \mathbb{R}$. For each $x \in \mathcal{X}$, the function $z \mapsto \mathcal{K}(z, x)$ belongs to the Hilbert space and we have the reproducing property:

$$\langle \mathcal{K}(\cdot, x), f \rangle_{\mathcal{H}} = f(x), \qquad \forall f \in \mathcal{H},$$

To simplify notation, we adopt the shorthand $\Phi_x = \mathcal{K}(\cdot, x)$.

In this paper, we need to study functions in the tensor product of two RKHSs. Suppose there exist two RKHSs. One is for state and action space $\mathcal{S} \times \mathcal{A}$ with inner product $\langle f, g \rangle_{\mathcal{H}_1}$ and kernel function $\mathcal{K}_1 : (\mathcal{S} \times \mathcal{A}) \times (\mathcal{S} \times \mathcal{A}) \to \mathbb{R}$, the other is for state space $\mathcal{S}$ with inner product $\langle f, g \rangle_{\mathcal{H}_2}$ and kernel function $\mathcal{K}_2 : \mathcal{S} \times \mathcal{S} \to \mathbb{R}$. Then $\mathcal{H} = \mathcal{H}_1 \otimes \mathcal{H}_2$ is an RKHS with reproducing kernel $\mathcal{K} = \mathcal{K}_1 \otimes \mathcal{K}_2 : (\mathcal{S} \times \mathcal{A} \times \mathcal{S})^2 \to \mathbb{R}$ (Berlinet & Thomas-Agnan, 2011), where

$$\mathcal{K}(((s_1, a_1), \tilde{s}_1), ((s_2, a_2), \tilde{s}_2)) = \mathcal{K}_1((s_1, a_1), (s_2, a_2))\mathcal{K}_2(\tilde{s}_1, \tilde{s}_2).$$

It has the reproducing property:

$$\langle \mathcal{K}(\cdot, ((s, a), \tilde{s})), M \rangle_{\mathcal{H}} = M((s, a), \tilde{s}), \qquad \forall M \in \mathcal{H}.$$

Furthermore, we equip $\mathcal{S} \times \mathcal{A}$ with measure $\mu_1$ and $\mathcal{S}$ with measure $\mu_2$. In the subsequent analysis, we assume $\mu_1$ and $\mu_2$ are Lebesgue measures with $\mu_1(\mathcal{S} \times \mathcal{A}) < \infty$ and $\mu_2(\mathcal{S}) < \infty$. This assumption is easy to satisfy since the state and action space are always compact sets.

**Assumption 1.** *(RKHS Embedding of Transition Model). For each $(s, a) \in \mathcal{S} \times \mathcal{A}$, $\tilde{s} \in \mathcal{S}$, reproducing kernel $\Phi_{s,a}(\cdot) = \mathcal{K}_1(\cdot, (s, a))$, $\Psi_{\tilde{s}}(\cdot) = \mathcal{K}_2(\cdot, \tilde{s})$ are given as a priori. There exists an unknown function $M^*((x, z), y) \in \mathcal{H} = \mathcal{H}_1 \otimes \mathcal{H}_2$ such that*

$$P(\tilde{s}|s, a) = \left\langle \iint_{\mathcal{S} \times \mathcal{A}} \Phi_{s,a}(x, z) M^*((x, z), \cdot) dx dz, \Psi_{\tilde{s}}(\cdot) \right\rangle_{\mathcal{H}_2}.$$

*Here, we call the function $M^*$ as a transition core.*

**Remark.** When considering the finite-dimensional transition core, Assumption 1 can be reduced to feature embedding (Yang & Wang, 2020). However, the finite-dimensional transition core is hard to capture nonlinear relationship. Assumption 1 can be satisfied if the coefficient of basis expansion for $P((s, a), \tilde{s})$ decays fast enough. In specific, let $\{a_{ij}\}$ be the coefficient of basis expansion $P((s, a), \tilde{s}) = \sum_{i,j} a_{ij} \phi_i(s, a) \psi_j(\tilde{s})$, where $\{\phi_i\}$ is the orthonormal basis in $\mathcal{L}^2(\mathcal{S} \times \mathcal{A})$ and $\{\psi_j\}$ is the orthonormal basis in $\mathcal{L}^2(\mathcal{S})$. Let $\{\gamma_i\}$ be the eigenvalues of $\mathcal{H}_1$ and $\{\mu_j\}$ be the eigenvalues of $\mathcal{H}_2$. Then Assumption 1 can be satisfied as long as $\sum_{i,j} a_{ij}^2 / \gamma_i^3 \mu_j < \infty$. For example, $\{\gamma_i\}$ and $\{\mu_j\}$ are polynomial decay, corresponding to $\mathcal{H}_1$ and $\mathcal{H}_2$ are sobolev spaces. Then Assumption 1 will be satisfied by most of the smooth functions. Details will be shown in Appendix D.

## 2.3 Generalization of Matrix Multiplication

In this paper, we generalize matrix multiplication from linear spaces to Hilbert spaces.

**Definition 2.** *(Generalization of matrix multiplication) $\mathcal{J}$ is a Hilbert space with inner product $\langle \cdot, \cdot \rangle_{\mathcal{J}}$. If $f(x, \cdot) \in \mathcal{J}$ and $g(\cdot, y) \in \mathcal{J}$, then $(f \circ g)(x, y)$ is defined as*

$$(f \circ g)(x, y) = \langle f(x, \cdot), g(\cdot, y) \rangle_{\mathcal{J}}.$$

**Remark.** It is consistent with matrix multiplication. Consider the case when $\mathcal{J} = \mathbb{R}^n$ with inner product $\langle a, b \rangle_2 = \sum_{k=1}^n a_k b_k$. Then for matrix $A, B$, we have $AB(i.j) = \sum_{k=1}^n a_{ik} b_{kj}$.

To clarify differences, we use operator $\circ$ when $\mathcal{J} = \mathcal{H}_2$ and use operator $\tilde{\circ}$ when $\mathcal{J} = \mathcal{L}^2$ with Lebesgue measures.

## 3 RL Exploration in RKHS

In this section, we study the way to balance exploration and exploitation given a set of reproducing kernels. Our goal is to develop an algorithm and give a regret bound.

### 3.1 Estimating the Transition Core

The high-level idea of the algorithm is to approximate the unknown transition core $M^*$ using the collected data. Suppose that at time step $t = (n, h)$, we observe the state-action-state transition triplet $(s_t, a_t, \tilde{s}_t)$, where $\tilde{s}_t := s_{t+1}$. For simplicity, we denote the associated reproducing kernels by

$$\Phi_t := \Phi_{(s_t, a_t)}(\cdot) = \mathcal{K}_1(\cdot, (s_t, a_t)) \quad \text{and} \quad \Psi_t := \Psi_{\tilde{s}_t}(\cdot) = \mathcal{K}_2(\cdot, \tilde{s}_t).$$

Let $K(x, y) = \int_{\mathcal{S}} \Psi_{\tilde{s}}(x) \Psi_{\tilde{s}}(y) d\tilde{s}$. We can prove that (see details in Appendix B)

$$\mathbb{E}\left[\Psi_{n,h}(v) | s_{n,h}, a_{n,h}\right] = \iint_{\mathcal{S} \times \mathcal{A}} \Phi_{n,h}(x, z)(M^* \circ K)((x, z), v) dx dz.$$

Denote our estimator of $M^*$ as $M_n$, then $M_n$ is the solution to the following ridge regression problem:

$$M_n = \arg\min_M \sum_{n' < n, h \leq H} \left\| \Psi_{n',h}(v) - \iint_{\mathcal{S} \times \mathcal{A}} \Phi_{n',h}(x, z)(M \circ K)((x, z), v) dx dz \right\|_2^2 + \lambda_n \|M\|_{\mathcal{H}}^2,$$

where $\lambda_n > 0$ is a user-defined regularization parameter. We solve the ridge regression problem by Fréchet derivative. After solving the ridge regression problem, we get $M_n$ satisfies the following equation (see details in Appendix C):

$$\sum_{n' < n, h \leq H} \mathcal{K}\tilde{\circ}\left[\Phi_{n',h} \otimes (\mathcal{K}_2\tilde{\circ}(\Psi_{n',h} - \Phi_{n',h}\tilde{\circ}(M_n \circ K)))\right] = \lambda_n M_n \tag{1}$$

For simplicity in the subsequent analysis, we define operator $A_n$ as

$$A_n(f) = \frac{1}{(n-1)H} \sum_{n' < n, h \leq H} \mathcal{K}\tilde{\circ}\left[\Phi_{n',h} \otimes (\mathcal{K}_2\tilde{\circ}(\Phi_{n',h}\tilde{\circ}(f \circ K)))\right]. \tag{2}$$

### 3.2 Upper Confidence RL in RKHS

In online RL, a critical step is to estimate future value of the current state and action use dynamic programming. To better balance exploitation and exploration, we want to use a confidence ball to construct an optimistic value function estimator. At episode $n$:

$$\forall (s, a) \in \mathcal{S} \times \mathcal{A}: \qquad Q_{n, H+1}(s, a) = 0 \quad \text{and ,} \tag{3}$$

$$\forall h \in [H]: \qquad Q_{n,h}(s, a) = r_h(s, a) + \max_{M \in B_n} \int_{\mathcal{S}} P_M(\tilde{s}|s, a) V_{n,h+1}(\tilde{s}) d\tilde{s}, \tag{4}$$

where

$$P_M(\tilde{s}|s, a) = \left\langle \iint_{\mathcal{S} \times \mathcal{A}} \Phi_{s,a}(x, z) M((x, z), \cdot) dx dz, \Psi_{\tilde{s}}(\cdot) \right\rangle_{\mathcal{H}_2}$$

and

$$V_{n,h}(s) = \max_a Q_{n,h}(s, a) \quad \forall s, a, n, h.$$

Here the confidence ball $B_n$ is constructed as

$$B_n := \left\{ M \in \mathcal{H} : \left\| A_n(M - M_n) + \frac{\lambda_n}{(n-1)H}(M - M_n) \right\|_2^2 \leq \beta_n \right\}, \tag{5}$$

where $\beta_n$ is a parameter to be determined later. In each step, at state $s_{n,h}$, we play action $a_{n,h} = \arg\max_a Q_{n,h}(s_{n,h}, a)$. The full algorithm is given in Algorithm 1.

---

**Algorithm 1:** Upper Confidence Reinforcement Learning in RKHS (RKHS-RL)

---

**Input:** An episodic MDP environment $\mathcal{M} = (\mathcal{S}, \mathcal{A}, P, r, H, s_0)$;
      Reproducing Kernels $\mathcal{K}_1 : (\mathcal{S} \times \mathcal{A}) \times (\mathcal{S} \times \mathcal{A}) \to \mathbb{R}$ and $\mathcal{K}_2 : \mathcal{S} \times \mathcal{S} \to \mathbb{R}$;
      Total number of episodes $N$;

1 **Initialize:** $A_1 = \mathcal{I} \in \mathcal{H}_1 \otimes \mathcal{H}_1, M_1 = 0 \in \mathcal{H}_1 \otimes \mathcal{H}_2$, where

$$\mathcal{I}\left((s,a),(s',a')\right) = \left\{ \begin{array}{l} 1, (s,a) = (s',a') \\ 0, \text{ otherwise} \end{array} \right. .$$

2 **for** *episode* $n = 1, 2, \cdots, N$ **do**
3      Let $\{Q_{n,h}\}$ be given in (3) and (4) using $M_n, \beta_n$;
4      **for** *stage* $h = 1, 2, \cdots, H$ **do**
5          Let the current stage be $s_{n,h}$;
6          Play action $a_{n,h} = \mathrm{argmax}_{a \in \mathcal{A}} Q_{n,h}(s_{n,h}, a)$;
7          Record the next state $s_{n,h+1}$;
8      **end**
9      Compute $A_{n+1}$ using (2);
10      Compute $M_{n+1}$ using (1);
11 **end**

---

**Remark.** We present a practical approach for addressing the maximum problem in equation 4. Leveraging the representative theorem, we express the objective function $M$ as a linear combination of kernels. By Lagrange duality, we can reformulate the confidence ball problem into a penalized version. Substituting the linear combination for $M$, we can reframe the original infinite-dimensional problem as a finite-dimensional quadratic counterpart. Consequently, this transformation enables an efficient solution to the problem.

### 3.3 REGRET BOUNDS

We first introduce some regularity conditions of the RKHS and the user-defined parameter $\lambda_n$.

**Assumption 2.** *(RKHS regularity) Let RKHSs and transition core satisfy the following conditions:*

1. $\|M\|_2^2 \leq C, \quad \|\mathcal{K}_1\|_2 \leq C \quad$ *and* $\quad \|\mathcal{K}_2\|_2 \leq C$;

2. $\forall (s,a) \in \mathcal{S} \times \mathcal{A}, \tilde{s} \in \mathcal{S}: \quad 0 < c \leq \|\Phi_{s,a}\|_2 \leq C \quad$ *and* $\quad \|\Psi_{\tilde{s}}\|_2 \leq C$;

3. $\forall f \in \mathcal{H}_2: \quad 0 < c\|f\|_2 \leq \|\mathcal{K}_2 \tilde{\circ} f\|_2 \leq C\|f\|_2$;

4. $\forall g \in \mathcal{H}: \quad c\|g\|_2 \leq \|\mathcal{K}\tilde{\circ}g\|_2 \quad$ *and* $\quad c\|g\|_2 \leq \|g\tilde{\circ}\mathcal{K}_2\|_2 \leq C\|g\|_2$.

**Remark.** Assumption 2 implies having an upper bound on the transition core $M$, as well as upper and lower bounds on reproducing kernels, ensuring that the eigenvalues of the operators are within a normal range. In the finite-dimensional case, RKHS-regularity reduces to feature regularity, as considered in (Yang & Wang, 2020).

**Assumption 3.** *(choice of $\lambda_n$) Let $\lambda_n > 0$ satisfies the following condition:*

$$\lambda_n \leq CH\sqrt{n \ln n}.$$

With the conditions above we can provide the regret bound.

**Theorem 1.** *Suppose Assumption 1,2 and 3 hold. After $T = NH$ steps, Algorithm 1 achieves regret bound:*

$$\mathrm{Regret}(T) \leq \tilde{\mathcal{O}}(H\sqrt{T}),$$

*if we let $\beta_n = \tilde{\mathcal{O}}(1/(nH))$.*

**Remark.** The regret bound implies $\mathrm{Regret}(T)/N \to 0$ when $N \to \infty$. Recall Definition 1, we can prove that the value function $V$ converges to the optimal value function $V^*$ as the episode $N$ increases. It ensures that RHKS-RL can learn the optimal policy for sufficiently large numbers

of $N$. The regret bound in Theorem 1 is of the same rate in terms of $T$ and tighter in terms of $H$ when compared with the regret bound $\mathcal{O}(H^2 d \log T \sqrt{T})$ for feature embedding case (Yang & Wang, 2020).

**Proof sketch.** The proof consists of two steps. (a) First, if the transition core $M^*$ is always inside the confidence ball $B_n$, the Q-functions provide optimistic estimates of the optimal values and we can obtain the regret bound using the sum of confidence bounds on the sample path. (b) Second, we show that with high probability, the requirement in (a) is satisfied.

### 3.4 RKHS-RL IN FINITE-DIMENSIONAL CASE

We utilize our model and algorithm to address a case involving a finite number of states and actions, which represents a straightforward scenario that our algorithm is well-suited to handle. In such circumstances, the finite-dimensional transition core proves adequate for resolving all conditions. Our objective is to implement and evaluate our algorithm in practical settings, comparing it with established techniques such as Q-Learning and Sarsa, in order to demonstrate its efficacy in achieving a lower regret.

**Establish finite dimensional RKHS $\mathbb{H}$.** Denote $\mathcal{X} = \{1, ..., p\}$ whose functional space is $\mathbb{R}^p$ and each point has the same measure. For any $\alpha, \beta \in \mathbb{R}^p$, define $\langle \alpha, \beta \rangle_{\mathbb{H}} = \alpha^\top \beta$ and $\Phi_x = (0, ...0, 1, 0, ...)^\top$, which equals 1 only at index $x$. Define the kernel as $K(x, y) = \Phi_x^\top \Phi_y$. It means $K(x, y) = 1$ if and only if $x = y$. Now we show that $\mathbb{H}$ is an RKHS. First, We have $\forall z \in \mathcal{X}, K(z, \cdot) = \Phi_z(\cdot) \in \mathbb{R}^p$. Then, we check reproducing property: $\langle K(\cdot, x), f(\cdot) \rangle_{\mathbb{H}} = \sum_y K(y, x) f(y) = f(x)$. So we can introduce RKHS into the finite-dimensional space with its inner product the same as the inner product in $\mathcal{L}^2$ with Lebesgue measures.

**Estimating the transition core.** Using the same method in Section 3.1, we can get a finite dimensional version of equation 1:

$$M_n = \Big( \sum_{n' < n, h \leq H} \Phi_{n',h} \Phi_{n',h}^\top + \lambda_n I \Big)^{-1} \Big( \sum_{n' < n, h \leq H} \Phi_{n',h} \Psi_{n',h}^\top \Big), \qquad (6)$$

**Closed-form Confidence Bounds** In the finite-dimensional case, the maximum in equation 4 can be expressed in a simpler form. By applying the Kronecker product, vectorization, and certain maximization methods, we can derive closed-form confidence bounds:

$$Q_{n,h}(s, a) = r_h(s, a) + \sqrt{\beta_n} \big\| V_{n,h+1} \otimes \omega_n^{-1} \Phi_{s,a} \big\|_2 + \Phi_{s,a}^\top M_n V_{n,h+1},$$

where

$$\omega_n = \frac{1}{(n-1)H} \Big( \sum_{n' < n, h \leq H} \Phi_{n',h} \Phi_{n',h}^\top + \lambda_n I \Big).$$

The full algorithm is given in Algorithm 2.

## 4 SIMULATION

We conduct simulations to assess the performance of finite-dimensional RKHS-RL. The experiment examines the asymptotic property of the average value (Figure 1a), which indicates that the solution of our function is stable. Additionally, we evaluate the regret bound proposed in our paper. As shown in Figure 1b, we observe that $\text{Regret}(T)/N^{1/2}$ is bounded. So the results align with Theorem 1.

**Set up of simulation.** We conduct an experiment with $|\mathcal{S}| = 20$, $|\mathcal{A}| = 4$, $N = 1000$, and $H = 8$. We set $\mathcal{S} = \{0, 1, \cdots, 19\}$ and $\mathcal{A} = \{0, 1, 2, 3\}$. We randomly generate transition probability $P$ and reward function $r$. We calculate the optimal value function $V^*$ by using the transition probability $P$ and reward function $r$. Then we use Algorithm 2 to calculate $V$. Regret, denoted as $\text{Regret}(T)$, is $V^* - V$. Regret over $N^{\frac{1}{2}}$ is represented as $\text{Regret}(T)/N^{\frac{1}{2}}$. We choose $\lambda_n = \sqrt{n}/160$. Figure 1(a) displays the curve of $(\sum_{n=1}^N \sum_{h=1}^H r_h(s_{n,h}, a_{n,h}))/N$ as $N$ increases. The curve's asymptotic convergence indicates that our algorithm's returned value is stable as $N$ becomes sufficiently large. In Figure 1(b), we depict our regret. The cumulative value function under the optimal policy is represented as cumulative $V^*$, while cumulative $V$ denotes the cumulative value function during $N$ loops under Algorithm 2. Cumulative $V$ over $\sqrt{N}$ should be bounded as $N$ tends to infinity based

---

**Algorithm 2:** Finite Dimensional RKHS-RL

---

**Input:** An episodic MDP environment $\mathcal{M} = (\mathcal{S}, \mathcal{A}, P, r, H, s_0)$;
    Total number of episodes $N$;

1 **Initialize:** $A_1 = I \in \mathbb{R}^{(|\mathcal{S}| \times |\mathcal{A}|)^2}, M_1 = 0 \in \mathbb{R}^{(|\mathcal{S}| \times |\mathcal{A}|) \times |\mathcal{S}|}$ ;

2 **for** *episode* $n = 1, 2, \cdots, N$ **do**

3   Let $\{Q_{n,h}\}$ be defined as follows:

   $\forall (s,a) \in \mathcal{S} \times \mathcal{A}: \quad Q_{n,H+1}(s,a) = 0 \quad$ and

4      $\forall h \in [H]: \quad Q_{n,h}(s,a) = r_h(s,a) + \sqrt{\beta_n} \big\| V_{n,h+1} \otimes \omega_n^{-1} \Phi_{s,a} \big\|_2 + \Phi_{s,a}^\top M_n V_{n,h+1}.$

  where

$$V_{n,h}(s) = \max_a Q_{n,h}(s,a) \quad \forall s,a,n,h$$

  and

$$\omega_n = \frac{1}{(n-1)H}\Big( \sum_{n' < n, h \leq H} \Phi_{n',h} \Phi_{n',h}^\top + \lambda_n I \Big)$$

  **for** *stage* $h = 1, 2, \cdots, H$ **do**

5    Let the current stage be $s_{n,h}$;

6    Play action $a_{n,h} = \text{argmax}_{a \in \mathcal{A}} Q_{n,h}(s_{n,h}, a)$;

7    Record the next state $s_{n,h+1}$;

8   **end**

9   Compute $M_{n+1}$ using equation 6

10 **end**

---

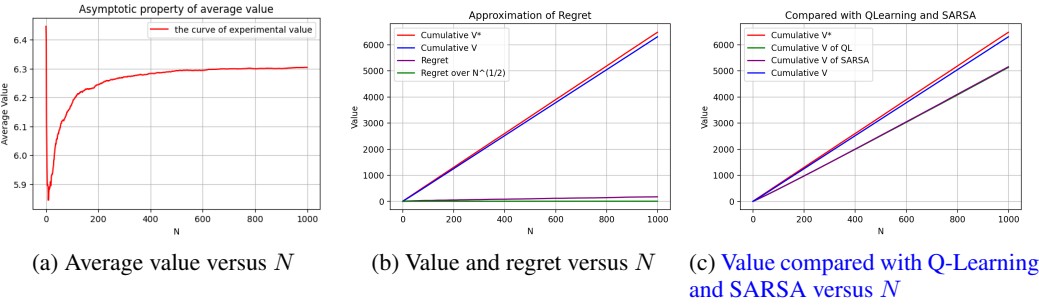

(a) Average value versus $N$  (b) Value and regret versus $N$  (c) Value compared with Q-Learning and SARSA versus $N$

Figure 1: Regrets and Value Function Estimation

on our theory. Figure 1(c) displays our cumulative value compared with Q-Learning and SARSA. Our algorithm is better than these in cumulative value, partly because their general explore policy is $\epsilon$-greedy exploration (where we choose $\epsilon = 0.9$ in our algorithm). So it could be understood that these two algorithms at most reaches probably $0.9V^*$.

## 5 SUMMARY

This paper proposes a model for online RL in the setting of RKHS. The core assumption is the RKHS-embedding of transition probability, which exhibits exceptional generalization capabilities and can handle nonlinearity. Based on this model, we provide a non-parametric algorithm, RKHS-RL, for solving episodic online RL problems. The RKHS-RL algorithm estimates the transition core using ridge regression and balances exploration and exploitation through infinite-dimensional confidence balls. We prove that RKHS-RL has the regret bound of $\tilde{\mathcal{O}}(H\sqrt{T})$, where $T$ represents the time step of the algorithm, and $H$ denotes the horizon of the MDP. It's a regret bound that is near-optimal in $T$, polynomial in $H$ and independent of $d$. We also apply RKHS-RL to the finite-dimensional case and derive a finite-dimensional version of the algorithm. However, it remains open that whether the RKHS regularity condition can be relaxed.

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
