# OpenReview forum: "Exploring the State and Action Space in Reinforcement Learning with Infinite-Dimensional Confidence Balls"
_ICLR.cc/2024/Conference — Submitted to ICLR 2024_

### Official Review · Reviewer_u1Ph · 2023-10-30

**Soundness:** 2 fair
**Presentation:** 2 fair
**Contribution:** 2 fair
**Rating:** 3
**Confidence:** 4

**Summary:**

This paper studies online RL where (1) both state and action spaces are assumed to be infinite and (2) the transition kernel is approximated by an RKHS. The paper proposes a model-based algorithm to directly optimize over the space of transition kernels and provides a concentration analysis for the concentration of the transition kernel. The resulting algorithm is shown to have $\tilde{O}(\sqrt{T})$ regret. Simulation studies are included.

**Strengths:**

- The paper proposes a possibly computationally efficient algorithm for RL with function approximation when state and action spaces are infinite.
- Assuming that the transition kernel itself is drawn from an RKHS seems an interesting idea that has not been explored fully in prior literature.

**Weaknesses:**

1. The proof appears to be incomplete and is hard to follow. It is not clear how the concentration analysis in A.2 helps the regret bound in A.1. It is not clear how these two parts relate to each other and it is further unclear how "the sum of the right-hand side" is bounded in Lemma 3, without invoking the later lemmas.
2. Some related works appear to be missing, such as earlier works on sample-efficient RL for low-rank MDPs or MDPs with bounded Bellman-Eluder dimension, or recent works such as admissible Bellman characterization.
3. Is it assumed, perhaps implicitly, that the reward function is known beforehand? The paper appears to estimate the transition kernel only and does not discuss how the reward function $r$ is estimated. It would be surprising if the paper can obtain the regret guarantee without any regularity assumptions on the reward function.
4. The algorithm design appears to be similar to earlier works on RL with general function approximation: it appears to be a specialization of earlier algorithm such as OLIVE for when (1) only the transition is estimated and (2) the function class is known to be an RKHS, which allows the optimization problem to be written directly over the space of $M$.
5. Algorithm 2 requires $S$ and $A$ to be finite, which is not assumed by Section 3.4.
6. The remark after Algorithm 1 is a bit misleading. Without any assumption on $r$, I am not sure if $\max_{a} Q(s, a)$ can be done efficiently.

**Questions:**

1. Can the authors provide more details on the proof?
2. Can the authors discuss how the paper relates to additional prior works not discussed in the paper?

---

> ### Author Response · Authors · 2023-11-21
>
> **Rebuttal:**
>
> **Q1:**
> The proof appears to be incomplete and is hard to follow. It is not clear how the concentration analysis in A.2 helps the regret bound in A.1. It is not clear how these two parts relate to each other and it is further unclear how "the sum of the right-hand side" is bounded in Lemma 3, without invoking the later lemmas.
>
> **A1:**
> We're really sorry if any process of our proof puzzled you. Let us explain it to you.
> After we get the Lemma 3, which means $Regret(NH) \lesssim \sqrt{\beta_{N+1}}NH^2+H\sum_{n=1}^{N}\mathbb{P}[E_n=0]$, it is natural for us to consider bounding the right side. The first term of the right side is determined by the radius of the confidence ball, while the second term of the right side require the probability that $M^*$ is in the confidence ball. It is actually a trade-off about the confidence ball, so we want to learn more about how "small" can the confidence ball be if we want to ensure a high probability that $M^*$ is in the confidence ball, which is the Lemma 6, the final goal of A.2 Concentration. So we start A.2.
>
> Then in the A.3, we use Lemma 6, which shows the relation between the two terms of the right side of Lemma 3, and set optimal values to minimize the order of the sum of the two terms. Finally, by Lemma 3, we complete the bound of the Regret.
>
> We don't exactly know if we have explained it clearly. So if there is still any problem, feel free to question us. What's more, thanks to your advice, we have added some detailed ideas of proof in our revised version.
>
> **Q2:**
> Some related works appear to be missing, such as earlier works on sample-efficient RL for low-rank MDPs or MDPs with bounded Bellman-Eluder dimension, or recent works such as admissible Bellman characterization.
>
> **A2:**
> Related work:
>
> There is a line of work on sample efficient RL in low rank MDPs [1][2]. [3] introduces Bellman Eluder dimension to find the minimal structure assumptions that empower sample efficient learning. [4] proposes an Admissible Bellman Characterization class that subsumes nearly all MDP models in the literature for tractable RL.
>
> [1]Agarwal A, Kakade S, Krishnamurthy A, et al. Flambe: Structural complexity and representation learning of low rank mdps[J]. Advances in neural information processing systems, 2020, 33: 20095-20107.
>
> [2]Shah D, Song D, Xu Z, et al. Sample efficient reinforcement learning via low-rank matrix estimation[J]. Advances in Neural Information Processing Systems, 2020, 33: 12092-12103.
>
> [3]Jin C, Liu Q, Miryoosefi S. Bellman eluder dimension: New rich classes of rl problems, and sample-efficient algorithms[J]. Advances in neural information processing systems, 2021, 34: 13406-13418.
>
> [4]Chen Z, Li C J, Yuan A, et al. A general framework for sample-efficient function approximation in reinforcement learning[J]. arXiv preprint arXiv:2209.15634, 2022.
>
> **Q3:**
> Is it assumed, perhaps implicitly, that the reward function is known beforehand? The paper appears to estimate the transition kernel only and does not discuss how the reward function $r$ is estimated. It would be surprising if the paper can obtain the regret guarantee without any regularity assumptions on the reward function.
>
> **A3:**
> In our paper, we need to know the immediate reward $r_h(s,a)$ after playing $a$ at $s$. This is in fact without loss of generality because learning about the environment $P$ is much harder than learning about $r$. In the case if $r$ is unknown, we can extend our algorithm by adding a step of optimistic reward estimation like in LinUCB. There are also works having the same assumptions on reward.[1][2][3]
>
> [1]Yang L, Wang M. Reinforcement learning in feature space: Matrix bandit, kernels, and regret bound[C]//International Conference on Machine Learning. PMLR, 2020: 10746-10756.
>
> [2]Agrawal S, Jia R. Optimistic posterior sampling for reinforcement learning: worst-case regret bounds[J]. Advances in Neural Information Processing Systems, 2017, 30.
>
> [3]Azar M G, Osband I, Munos R. Minimax regret bounds for reinforcement learning[C]//International Conference on Machine Learning. PMLR, 2017: 263-272.
>
> **Q4:**
> Algorithm 2 requires $S$ and $A$ to be finite, which is not assumed by Section 3.4.
>
> **A4:**
> The simulation (Algorithm 2) is based on a special case that the state and action space is finite (and thus the kernel space must be finite-dimensional) just as the Section 3.4, where we have defined the setting of the RKHS model in this specific case.

---

> > ### Comment · Reviewer_u1Ph · 2023-11-22
> >
> > Thank the authors for their response.
> >
> > - I haven't found the revision that the authors mentioned. Could you double check to see if it has been updated?
> > - Details behind the proof is still missing. My original question is on how the bound in Lemma 3 can be *obtained* using the previous results. Usually this is done by some variant of elliptical potential growth lemma, which I cannot find in the proof. Even then, assuming the proof is correct, I still believe the proof to be poorly organized, and much more care need to go into writing and presenting the proof for a top tier conference such as ICLR.
> > - I don't think the relationship between the proposed algorithm and earlier results such as GOLF or OLIVE has been discussed sufficiently. How is the proposed approach different from simply adapting these earlier results to the RKHS setting?
> > - The assumption that the reward is known is not mentioned in the paper. Earlier works, including those cited, explicitly made this assumption clear. Even then, the assumption is not without loss of generality: we simply do not have polynomial-sample confidence bounds for all possible choices of $r$ (e.g. $r$ is nonparameteric and belongs to a function class with exponential complexity).
> > - Algorithm 1 cannot be done efficiently when $r(s, a)$ is nonconcave / nonconvex in $a$ and the action space is continuous.

---

> > > ### Author Response · Authors · 2023-11-23
> > >
> > > 1. Thank you. We've now updated the revision. If you have questions about the subsequent answers, you could refer to the revised version.
> > >
> > > 2. The elliptical potential lemma in [1][2]: $\sum_{t=1}^TA_t^T\Sigma_tA_t \leq 2log\frac{det\Sigma_1}{det\Sigma_{T+1}} \leq 2dlog(1+\frac{T}{\lambda d})$, is in a finite-dimensional background ($A$ is a finite-dimensional vector) and use finite-dimensional techniques. Therefore, it can be used in Yang \& Wang 2020 to bound the term $\mathbf{E}[\sum_{n=1}^N\sum_{h=1}^H \sqrt{min(1,\omega_{n,h}^2)}]$. \\
> > > However, it cannot be used in our proof directly, since our lemma is based on infinite-dimensional vector space. So we need to use Assumption 2 and the choice of $\lambda_n$ to bound $\mathbf{E}[\sum_{n=1}^N\sum_{h=1}^H\omega_{n,h}]$. (Remark: we need $\lambda_n$ to be $o(nH)$, which can be satisfied as we choose $\lambda_n = \tilde{\mathcal{O}} (H\sqrt{n})$.) What's more, Assumption 2 (RKHS regularity) is not very strong, since if we reduce it to the finite-dimensional space, condition 1,2 are always satisfied and condition 3,4 can be satisfied if the kernel is invertible and the corresponding constant is related to its eigenvalue.
> > >
> > > [1].Nima Hamidi, Mohsen Bayati The Elliptical Potential Lemma for General Distributions with an Application to Linear Thompson Sampling 	arXiv:2102.07987 [stat.ML]
> > >
> > > [2].Alexandra Carpentier, Claire Vernade, Yasin Abbasi-Yadkori The Elliptical Potential Lemma Revisited arXiv:2010.10182 [stat.ML]
> > >
> > > 3. OLIVE eliminates functions that have high average Bellman error under
> > > a certain policy. The idea behind OLIVE is similar to the confidence ball since in this case only the transition policy needs to be estimated. However, OLIVE focuses on PAC learning and only gives the sample complexity. We provide regret bound which means the regret of each step is important.
> > >
> > >
> > > 4. Thanks for your advice. We have clarified that our reward function should be known in advance in our revised version. Surely, we do not have polynomial-sample confidence bounds for all possible choices of reward function. But if the reward function is unknown and in certain conditions like the Linear MDP model and the Linear Mixture MDP model, this reward function can only generate a lower order regret compared with the transition core.
> > >
> > > 5. In our algorithm, we suppose that we know the reward function in advance, so the Algorithm 1 can be seen as an optimization problem of the function $f(M)$, considering the Ball is a convex set and the updating Q is actually a linear function of $M$, this algorithm could be done efficiently. What's more, the optimization problem of $M$ is irrelevant of the reward function.

---

### Official Review · Reviewer_oU2p · 2023-10-31

**Soundness:** 3 good
**Presentation:** 3 good
**Contribution:** 3 good
**Rating:** 6
**Confidence:** 3

**Summary:**

This paper introduces a model-based algorithm for episodic MDP whose transition probability is embedded in a given reproducing kernel Hilbert space. The assumption of RKHS-embedding of transition probability can handle non-linear relationships in transition probability. The proposed algorithm (RKHS-RL) estimates the transition core using ridge regression based on the collected data, and constructs an optimistic action-value function based on the infinite dimensional confidence ball. The proposed method achieves a dimension independent regret bound of $O(H \sqrt{T})$ where $H$ is the horizon length, $T$ is the total number of interaction between the agent and the environment. Furthermore, the authors confirm the performance of finite-dimensional RKHS-RL through experiments in a simple tabular setting.

**Strengths:**

- The motivation for the problem addressed in this paper is well-explained, drawing from the literature and related work. Additionally, the organization of the paper is appropriately structured to facilitate understanding.

- The RKHS-embedding of transition probability is useful in modeling non-linear transition probabilities for state-action pairs. The author has demonstrated that the regret bound of the proposed method can achieve dimension-free sub-linear regret.

**Weaknesses:**

- The computational complexity of the proposed method has not been addressed. In “Introduction”, the authors mentioned that the proposed model can handle infinite state and action spaces. However, it seems that the computation in Algorithms 1 and 2 is heavily influenced by the size of the state-action space. It would be helpful to specify how efficient the proposed method is both statistically and computationally compared to previous algorithms, particularly for infinite state spaces.

- The proposed algorithm appears similar to KernelMatrixRL (Yang & Wang, 2020). While the authors mentioned that the RKHS-embedding setting poses significant challenges compared to approach of Yang & Wang (2020), it would be beneficial to explain in detail what specific challenges arise and how they were addressed using mathematical techniques in the paper.

- There is a gap between the settings discussed in the paper and the simulations. The proposed method can handle settings that previous approaches, such as tabular MDPs, linear MDPs, and parametric MDPs, cannot. However the experiments were conducted in a simple tabular setting. Additionally, it is anticipated that Q-learning or SARSA would perform significantly better in the current experiments. It would be valuable to include comparisons with other baseline algorithms.

**Questions:**

1. How can the proposed method be applied to a continuous state-action space? The current planning approach seems challenging to implement in a continuous state-action space.

2. How was it possible to achieve a tight regret bound with respect to H when compared to the regret bound of Yang & Wang (2020)?

3. The regret bound of RKHS-RL does not include the effective dimension of the kernel. In that case, does the regret not depend on dimension even when using a finite-dimensional kernel? What is the key technique to eliminate dimension dependence when compared to KernelMatrixRL in Yang & Wang (2020) ?

4. In the finite-dimensional case of RKHS-RL, how is the regularity bound in Assumption 2 defined? Can we assume a bound independent of the size of the state-action space in finite-dimensional case?

5. Why does Figure 1-(b) show negative values for regret?

6. How does the following inequality in the Proof of Lemma 2 hold?
: $$ H || \Phi\_{n,h} \tilde{\circ} (M^* - M'\_n) ||_1 \le H || \Phi\_{n,h} \tilde{\circ} (M^* - M'\_n) ||_2 $$

---

> ### Author Response · Authors · 2023-11-21
>
> **Rebuttal:**
>
> **Q1:**
> The computational complexity of the proposed method has not been addressed.
>
> **A1:**
> Our algorithm and computation does not scale with the very large dimension of the state and action space, instead it is actually related to the Hilbert space which contains real-valued functions (that we want to include) based on the state and action space. For example, if there is a really large state and action space with a finite-dimensional ($d$) feature space, then our algorithm may only scale with the dimension $d$ hidden in the constant of our RKHS regularity. This case reduces to Yang & Wang. Therefore,  in the case of infinite state spaces, if the feature space is finite-dimensional, our algorithm coincides with Yang & Wang(2020), achieving an algorithm similar to our Algorithm 2.
>
> **Q2:**
> It would be beneficial to explain in detail what specific challenges arise and how they were addressed using mathematical techniques in the paper compared with Yang & Wang 2020.
>
> **A2:**
> Obtaining a similar outcome of regret bound in an infinite-dimensional case is not trivial. On one hand, many of the techniques available in the finite-dimensional setting are difficult to use in our proof. We overcome these problems using Fr$\acute{e}$chet derivatives, Azuma-type inequality for Banach space-valued martingales, and many other tools.
>
> On the other hand, generalizing the model to the RKHS framework is not a trivial task. Specifically, instead of naively assuming that the transition probability $P$ belongs to an RKHS, we assume a more delicate structure of $P$ which is that $P$ has a decomposable structure, as illustrated in Assumption 1. Only on this assumption can we derive an efficient algorithm to conduct RL but still make the model quite general as long as $P$ is smooth enough. Naively assuming $P$ belongs to an RKHS will lead to an unattainable outcome.
>
> **Q3:**
> There is a gap between the settings discussed in the paper and the simulations.
>
> **A3:**
> Our simulation is just a special case of our theory and Algorithm 1. Our theory and Algorithm 1 are much more universal and applicable to more cases since their modeling is infinite-dimensional. We have compared Algorithm 2 with established techniques such as Q-Learning and SARSA and found that our Algorithm 2 reaches a lower regret in this specific case in our latest edition.
>
> **Q4:**
> The regret bound of RKHS-RL does not include the effective dimension of the kernel. In that case, does the regret not depend on dimension even when using a finite-dimensional kernel?
>
> **A4:**
> The regret may depend on dimension when using a finite-dimensional kernel since constants in Assumption 2 (RKHS regularity) may depend on the dimension of the kernel. We can see A5 as an example.
>
> **Q5:**
> In the finite-dimensional case of RKHS-RL, how is the regularity bound in Assumption 2 defined?
>
> **A5:**
> Under our finite-dimensional case of RKHS-RL, Assumption 2 reduces to $\sum_{s,a,\widetilde{s}} M((s,a),\widetilde{s})^2 \leq C $, $d_1+d_2 \leq C$, $d_1\leq C$, $0\leq c\leq 1\leq C$, where $d_1$ is the dimension of $\mathcal{S}$ and $d_2$ is the dimension of $\mathcal{A}$. Our regularity bound isn't related to the dimension of state-action space but may scale with the dimension of the kernel (this case reduces to Yang & Wang).
>
> **Q6:**
> Why does Figure 1-(b) show negative values for regret?
>
> **A6:**
> Thanks for your correction. We made some mistakes in the former edition, mistaking the step-by-step strategy obtained by Q-learning as the best strategy when calculating the regret. We have corrected this mistake in the revised edition and added some comparisons with other algorithms like Q-learning and SARSA.
>
> **Q7:**
> How does the following inequality in the Proof of Lemma 2 hold? :
>
> **A7:**
> Thanks for your question. Under the assumption that the Lebesgue measure of space $S$ is finite, we can use the Jensen's inequality and get that
>
> $H ||$ $\Phi_{n,h}$ $\tilde{\circ}$ $(M^* - M'_n)$ $||_1$
>
> $\lesssim$  $H ||$ $\Phi_{n,h}$ $\tilde{\circ}$ $(M^* - M'_n)$ $||_2 $
>
> **Q8.**
> How was it possible to achieve a tight regret bound with respect to $H$ when compared to the regret bound of Yang & Wang (2020)?
>
> **A8.**
> We achieve a tight regret bound with respect to $H$ when compared to the regret bound of Yang \& Wang (2020) because the "radius" of our confidence ball $\beta_n$ converges faster than Yang \& Wang (2020).

---

> > ### Comment · Reviewer_oU2p · 2023-11-23
> >
> > Thanks for your response and I have no further questions.

---

### Official Review · Reviewer_3vbX · 2023-11-04

**Soundness:** 3 good
**Presentation:** 4 excellent
**Contribution:** 2 fair
**Rating:** 6
**Confidence:** 4

**Summary:**

This paper introduces a non-parametric online RL algorithm called RKHS-RL that overcomes the curse of dimensionality in RL by utilizing reproducing kernels and the RKHS-embedding assumption. The proposed algorithm can handle both finite and infinite state and action spaces, as well as nonlinear relationships in transition probabilities. The paper provides theoretical guarantees, demonstrating that RKHS-RL achieves a sublinear regret bound, making it an effective approach for RL problems.

**Strengths:**

The key contributions of the paper are:

1. **Theoretical Foundation**: It establishes a solid theoretical foundation for applying RKHS to reinforcement learning, providing a new perspective on how to handle the curse of dimensionality in such problems.

2. **Regret Bounds**: The paper presents a significant theoretical result by proving that RKHS-RL achieves sublinear regret bounds, specifically \( \tilde{O}(H\sqrt{T}) \), where \( T \) is the time step and \( H \) is the horizon of the Markov Decision Process (MDP). This indicates that the algorithm is efficient in balancing exploration and exploitation over time.

3. **Experimental evaluation**: The paper evaluates the performance of finite-dimensional RKHS-RL through simulations. The experiment examines the asymptotic property of the average value, which indicates that the solution of the function is stable. Additionally, the regret bound proposed in the paper is evaluated. The results show that the regret is bounded and align with Theorem 1.

**Weaknesses:**

1. **Empirical Evidence**: The paper could be strengthened by including more empirical evidence to support the theoretical findings. This includes detailed comparisons with existing methods, such as those mentioned in the references, to demonstrate the practical effectiveness of RKHS-RL.

2. **Scalability and Computation**: While the theoretical aspects are strong, the paper does not thoroughly address the scalability of the algorithm, especially considering the potential growth of the kernel matrix, which is a known issue in kernel methods as the number of state-action pairs increases. The paper could address potential computational bottlenecks more thoroughly, especially when scaling to very large state and action spaces.
The practicality of infinite-dimensional confidence balls in real-world applications is not addressed. The paper could improve by discussing how this aspect of the algorithm translates to practical implementations and what trade-offs might be involved.

3. **Generalization and Application**: The paper would benefit from a discussion on the generalization capabilities of RKHS-RL across different domains and a demonstration of its application to real-world problems, which are areas of interest in the references.

**Questions:**

0. **Typo**: Is it a typo in section 4 simulation?
> we observe that $\operatorname{Regret}(T) / N^{2 / 3}$ is bounded.
1. **Assumptions of RKHS-Embedding**:
Could you elaborate on the conditions under which the RKHS-embedding of transition probabilities is a valid assumption? Are there known classes of RL problems where this assumption may not hold?
2. **Algorithmic Scalability**:
How does the RKHS-RL algorithm scale with the dimensionality of the state and action spaces in practice? Are there computational constraints that could limit its application to large-scale problems?
3. **Comparison with Existing Methods**:
The paper would benefit from a comparative analysis with other RL algorithms. Have you conducted such comparisons, and if so, could you share these results?
4. **Hyperparameter Sensitivity**:
How sensitive is the RKHS-RL algorithm to the choice of hyperparameters, including the selection of kernels and regularization parameters in ridge regression?
5. **Practical Implementation**:
Can you provide insights into the practical implementation of infinite-dimensional confidence balls? How does this concept translate into a computationally feasible algorithm?

---

> ### Author Response · Authors · 2023-11-21
>
> **Rebuttal**:
>
> Thanks for your encouraging words and constructive comments. We sincerely appreciate your time in evaluating our work. Our point-to-point responses to your comments are given below.
>
> **Q1:**
> Empirical Evidence: The paper could be strengthened by including more empirical evidence to support the theoretical findings.
>
> **A1:**
> We have added the comparison with existing methods like Q-Learning and SARSA in our simulation of finite-dimensional case(Algorithm 2) and concluded that our algorithm achieves a lower regret.
>
> **Q2:**
> Scalability and Computation: While the theoretical aspects are strong, the paper does not thoroughly address the scalability of the algorithm. The paper could improve by discussing how this aspect of the algorithm translates to practical implementations and what trade-offs might be involved.
>
> **A2:**
> About Scalability: Since our algorithm needs to get the estimated function $M_n$ by the idea of kernel ridge regression, the computational complexity scales with the state-action pairs. However, it does not scale with the very large dimension of the state and action space. Instead, it is related to the Hilbert space which contains real-valued functions (that we want to study) based on the state and action space. For example, if there is a very large state and action space with finite features, then our algorithm may only scale with the feature dimension $d$ hidden in the constant of our RKHS regularity. This case reduces to Yang \& Wang.
> About Practicality: The practical implementation of the infinite-dimensional confidence ball is solving the maximum problem in equation (4). We present a practical approach for addressing the maximum problem. Leveraging the representer theorem, we express the objective function $M_n$ as a linear combination of kernels. Through the application of Lagrange Duality, we effectuate a conversion of the constrained optimization problem into a penalized version. By substituting the linear combination for $M_n$, we successfully reframe the original infinite-dimensional problem as a finite-dimensional quadratic counterpart. Consequently, this transformation enables an efficient solution to the problem.
>
> **Q3:**
> Generalization and Application: The paper would benefit from a discussion on the generalization capabilities of RKHS-RL across different domains and a demonstration of its application to real-world problems, which are areas of interest in the references.
>
> **A3:**
> Our algorithm can be used in different domain of state by using different kernels. In practice, our infinite-dimensional modelling can capture nonlinearity which is more practical. What's more, RKHS-based model can tackle the curse of dimensionality and it offers greater flexibility in capturing intricate patterns and relationships. Furthermore, kernel methods can proceed with high-dimensional data.
>
> **Q4:**
> Is it a typo in section 4 simulation? (we observe that $\operatorname{Regret}(T) / N^{2 / 3}$ is bounded.)
>
> **A4:**
> Thanks for your correction. We have corrected this typo in our revised version.
>
> **Q5:**
> Assumptions of RKHS-Embedding: Could you elaborate on the conditions under which the RKHS-embedding of transition probabilities is a valid assumption? Are there known classes of RL problems where this assumption may not hold?
>
> **A5:**
> Assumption 1 can be satisfied if the coefficient of basis expansion for $P((s,a),\tilde{s})$ decays fast enough. In specific, let $\{a_{ij}\}$ be the coefficient of basis expansion  $P((s,a),\tilde{s})=\sum_{i,j}a_{ij}h_i(s,a)g_j(\tilde{s})$, where $\{g_i\}$ is the orthonormal basis in $\mathcal{L}^2(\mathcal{S})$ and $\{h_i\}$ is the orthonormal basis in $\mathcal{L}^2(\mathcal{S}\times\mathcal{A})$. Let $\{\gamma_i\}$ be the eigenvalues of
> $\mathcal{H}_1$ and $\{\mu_j\}$ be the eigenvalues of $\mathcal{H}_2$.
>
> Then Assumption 1 can be satisfied as long as $\sum_{i,j} a_{ij}^2/\gamma_i^3 \mu_j < \infty$.
> For example, $\{\gamma_i\}$ and $\{\mu_j\}$ are polynomial decay,
> corresponding to $\mathcal{H}_1$ and $\mathcal{H}_2$ are sobolev spaces.
>
> $\gamma_i\leq i^{-\alpha}$ and $\mu_j\leq j^{-\beta}$ for some $\alpha$ and $\beta$. If $a_{ij}\leq i^{-3\alpha/2-1}j^{-\beta/2-1}$, then Assumption 1 will be satisfied. This rate of decay can be satisfied by most of the smooth functions. Details will be in the updated paper.
>
> **Q6:**
> Hyperparameter Sensitivity: How sensitive is the RKHS-RL algorithm to the choice of hyperparameters, including the selection of kernels and regularization parameters in ridge regression?
>
> **A6:**
> The choice of some hyperparameters like $\lambda_n$ is determined in our theoretical proof. Also, we have done some tests on the choice of $\lambda_n$ in simulation in our updated supplementary material. We can see that in this case, the order of $\lambda_n$ is insensitive to the regret. Considering the constant coefficient term, we can get it through cross-validation.
> 	The selection of kernel is determined by real-world setting.

---

### Author Response · Authors · 2023-11-21

**Rebuttal:**

Thanks to all the reviewers. We sincerely appreciate your time in evaluating our work.

Thanks to your advise, we have revised our article and our revisions are in blue. Besides some typos, we have added remarks, extra conditions, detailed ideas in the proof and comparisons with Q-Learning and SARSA in the simulation.

What's more, we want to answer some general questions. In the setting of our main theorem, the state and action space can be a space with finite measure, either continuous or discrete. The Hilbert space which contains real-valued functions (that we want to study) based on the state and action space is actually the crucial part in our theory. In our theory, the dimension of this Hilbert space (RKHS) could be infinite, while (Yang & Wang, 2020) just covered the finite-dimensional Hilbert space. The infinite-dimensional transition core space could cover nonlinear relations, and thus largely expand the potential space of the probability transition function. Furthermore, it is not trivial to obtain a similar outcome of regret bound in an infinite-dimensional case, both in proof and algorithm.

In the Section 3.4 of our article, we want to utilize our model and algorithm to address a case involving a finite number of states and actions, since in this case, we can cover all probability transition function without extra setting. But this is just a really simple case of our Algorithm 1, and does not reflect the theoretical breakthrough in our paper. In the simulation, we compare our result with Q-Learning and SARSA, and conclude that in this special case, our algorithm does better in cumulative value.

**About the practical implementation:**
The practical implementation of infinite-dimensional confidence ball is solving the maximum problem in equation (4). We present a practical approach for addressing the maximum problem. Leveraging the representer theorem, we express the objective function $M_n$ as a linear combination of kernels. Through the application of Lagrange Duality, we effectuate a conversion of the constrained optimization problem into a penalized version. By substituting the linear combination for $M_n$, we successfully reframe the original infinite-dimensional problem as a finite-dimensional quadratic counterpart. Consequently, this transformation enables an efficient solution to the problem.

---

### Meta-Review · Area_Chair_NQwf · 2023-12-08

**Metareview:**

The paper derives RL regret bounds with infinite dimensional RKHS. Although RKHS have been studied in RL before, this paper does so under less restrictive assumptions than prior art. They obtain regret bounds, and also perform very limited experiments.
Thus, this constitutes an interesting first steps towards provably efficient RL with infinite dimensional RKHS.
Nonetheless, the empirical evaluation is not convincing, and the key value of the paper is also questionable given the existing sizable literature about RL exploration. Finally, certain steps in the proofs appear to be missing and hard to follow. I encourage the authors to submit a work as complete as possible, in a way that makes it easy to verify the theorems, and with the relevant comparison with the literature, such as those suggested by the reviewers.

**Justification For Why Not Higher Score:**

The paper is not particularly groundbreaking

**Justification For Why Not Lower Score:**

N / A

---

### Decision · Program_Chairs · 2024-01-16

Reject